

# Dynamical mean field theory
# for models of confluent tissues and beyond

**Persia Jana Kamali and Pierfrancesco Urbani⋆**

Université Paris-Saclay, CNRS, CEA, Institut de physique théorique,
F-91191 Gif-sur-Yvette, France

⋆ pierfrancesco.urbani@cea.fr

## Abstract

We consider a recently proposed model to understand the rigidity transition in confluent tissues and we derive the dynamical mean field theory (DMFT) equations that describes several types of dynamics of the model in the thermodynamic limit: gradient descent, thermal Langevin noise and active drive. In particular we focus on gradient descent dynamics and we integrate numerically the corresponding DMFT equations. In this case we show that gradient descent is blind to the zero temperature replica symmetry breaking (RSB) transition point. This means that, even if the Gibbs measure in the zero temperature limit displays RSB, this algorithm is able to find its way to a zero energy configuration. We include a discussion on possible extensions of the DMFT derivation to study problems rooted in high-dimensional regression and optimization via the square loss function.

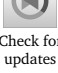

# 1 Introduction

Confluent tissues are a subclass of biological tissues for which, to a first approximation, cells tesselate the space, the simplest example being the epithelial tissue. In recent years there has been a huge effort to try to understand their physical properties starting from simple statistical mechanics models to be compared with real data in the field. The interest in this research line is due to many reasons including: (i) understanding morphogenesis and the interplay between the mechanics of stress propagation in aggregates of cells and the expression of chemical signals such as growth factors, and (ii) the dynamics of tumor growth and metastasis generation in healthy tissues. The two settings seem quite different but in fact they are rather similar. At the biochemical level, cells express growth factors through some metabolic pathway that is influenced by the current biochemical status. The expression of growth factors triggers growth in the assemblies of cells and this feedbacks into the expression of the growth factors themselves. If the process is properly balanced, one has a growing healthy tissue. Conversely, when cells are damaged, such mechanism is broken along the metabolic pathway. One crucial point is that in healthy tissues, biochemical signals may depend on the mechanics of the tissue itself. In other words stress relaxation and propagation in the aggregate of cells is fed back somehow into the metabolic pathway and the mechanical properties of single cells are crucial to determine the collective behavior. In tumor tissues such feedback loop is not controlled anymore and one has uncontrolled disordered growth.

It is therefore interesting to control the mechanical, elastic and plastic properties of assemblies of cells. From a statistical mechanics perspective, the simplest way to investigate this problem is by developing a simple set of models of confluent tissues. In recent years, Vertex and Voronoi models have been extensively studied [1–11]. In both types of models, a procedure to tessellate the space is introduced to properly define the cells which are therefore identified as the tiles of a geometric structure. Metabolic reasons are then invoked to constrain cells' shape and this is enforced with a proper cost function. A way to control cells' shape is by penalizing their volume and surface if they are away from the target values and this is done by using a square loss function in which deviations from the target shape have a cost that scales with the square of the actual deviation. A key problem is then to understand what are the dynamical properties of assemblies of cells moving under a set of both thermal and non-equilibrium forces and subject to the interactions inherited from the cost function. A typical setting consists in giving self-propulsion to the cells and then looking at whether cells can diffuse or not depending on the allowed target shape. In recent years it has indeed been shown that the target shape acts as a control parameter that can tune the rigidity of confluent tissues [1–11]. Enforcing a compact shape produces a glassy, solid tissue, while when the shape is loose, one gets a liquid tissue. However the phase boundary between these regions as well as the out-of-equilibrium phase transition between them is not well understood and therefore one needs to develop simple mean field models which can bring with their exact solubility, some analytical understanding.

A totally different line of research in statistical mechanics is focusing on the properties of artificial neural networks [12–14]. A simple way to think about them is as devices performing a non-linear regression task. In a huge number of cases, networks are trained such that their output to a given input pattern of a dataset, is the desired one (the label of an image for example). Therefore they can be thought to be doing regression tasks, which can be achieved with many cost functions, the simplest one being the square loss. In this case for each point in the dataset, the cost associated to a configuration of the network which does not provide the right label for that point, is defined as the square between the current output and the expected one. Therefore training the network corresponds to finding the set of parameters such that for each data point the network provides the right label.

Both models of confluent tissues as well as non-linear regression in high dimension can be thought as high-dimensional systems of non-linear equations (equality constraints) for the degrees of freedom (positions of cells in confluent tissues and the weights of the network in machine learning).

In [15] a simple model for a continuous constraint satisfaction problem with equality constraints has been studied and it has been shown to provide a similar phase diagram to the one of models of confluent tissues. The model has a rigidity transition at zero temperature which separates a satisfiable phase where all constraints can be satisfied and an unsatisfiable phase where all are violated. For confluent tissues, the satisfiable phase represents the liquid phase: one can move the degrees of freedom and stay at zero energy at the bottom of a *canyon landscape* [16]. Instead the unsatisfiable phase is glassy and the landscape is a high-dimensional rough arrangement of minima and saddles. The liquid phase in the regression setting denotes the overparametrized phase and the rigidity transition is nothing but the interpolation point in which the network is becoming too small to fit all dataset. The same picture holds for classification tasks and a parallel with the jamming transition of particles has been done in recent years [17–20].

The main purpose of this work is (i) to describe the solution of the infinite dimensional dynamics of the class of models described in [15] and (ii) to discuss how to solve numerically the corresponding equations. We will detail the derivation and the structure of the dynamical mean field theory (DMFT) equations [21] describing the dynamics in the infinite size limit. Interestingly we will show that the structure of the DMFT equations can be considered as an intermediate case between models where one gets a self-consistent stochastic process whose memory and noise kernels must be obtained numerically by sampling the process itself, see for example [22–25], and models for which the stochastic process can be integrated exactly to get the equations for the self-energy in the dynamical Dyson equations (for example the $p$-spin glass model [26, 27]).

This work is organized as follows: in Sec. 2 we recall the definition of the model and define the dynamical equations we are interested in. In Sec. 3 we construct the dynamical mean field theory (DMFT) that tracks the dynamics of the model in the thermodynamic limit. We highlight the similarities and differences with other DMFT analyses in other similar models and in Sec. 4 we describe the numerical procedure to solve the DMFT equations. In Sec. 5 we report the validation of the DMFT equations and some results we can get by integrating them numerically. Finally we conclude with some perspective and possible extensions of our analysis.

## 2 The model and dynamical equations

### 2.1 The model

The model considered in [15] is constructed out of an $N$-dimensional real vector $\underline{x} = \{x_1, \ldots, x_N\}$ constrained on the hypersphere $|\underline{x}|^2 = N$ which represents the phase space of the problem. This vector denotes the degrees of freedom that we are allowed to change in order to satisfy the constraints. They model the position of the centers of the cells in the Voronoi/Vertex models, and the weights to be adjusted to perform the non-linear regression task in the machine learning setting. In order to define a set of non-linear equality constraints, we consider a set of $M = \alpha N$ random matrices:

$$J_{ij}^{\mu} = J_{ji}^{\mu} \quad \mu = 1, \ldots M, \tag{1}$$

whose entries are Gaussian random variables with zero mean and unit variance. We then construct a set of gap variables $h_\mu$, one for each constraint:

$$h_\mu(\underline{x}) = \frac{1}{N} \sum_{i<j} J_{ij}^\mu x_i x_j \,. \tag{2}$$

From the gap variables one defines a set of $M$ equality constraints as

$$h_\mu(\underline{x}) = p_0 \qquad \forall \mu = 1, \ldots, M \,, \tag{3}$$

and $p_0 \geq 0$ plays the role of target shape in the case of confluent tissue models while it is a simple control parameter in the non-linear regression task. The square loss is then defined as the following Hamiltonian

$$H[\underline{x}] = \frac{1}{2} \sum_{\mu=1}^M \big(h_\mu(\underline{x}) - p_0\big)^2 \,. \tag{4}$$

The parameter $\alpha$ is a control parameter and tunes how much the degrees of freedom are constrained. If $\alpha$ is large, the model is expected to be in the UN-SAT/solid/underparametrized phase, while if $\alpha$ is small enough the model is expected to be in the SAT/liquid/overparametrized phase. The same scenario holds as a function of $p_0$ at fixed $\alpha$: at small $p_0$ the model is in the SAT phase while at large $p_0$ it is UNSAT. In [15] this model was studied at fixed $\alpha < 1$ as a function of $p_0$ mostly focusing on the properties of the Gibbs measure, as constructed from Eq. (4). In particular it has been shown that there exist a critical value $p_J$ which separates a region where, with probability one, a typical configuration of the zero temperature Gibbs measure has zero energy (and therefore all constraints are satisfied) from a phase where the ground state of $H$ has a positive energy. Therefore the value of $p_J$ represent the *thermodynamic* rigidity transition point in the context of confluent tissue modelling. Furthermore, in [15] it has been shown that the SAT phase is actually composed by two phases: for $p < p_G < p_J$ the Gibbs measure is replica symmetric and therefore one expects that simple local exploration algorithms will be able to equilibrate the zero energy manifold in phase space. Conversely, for $p \in [p_G, p_J]$ replica symmetry is broken and therefore the Gibbs measure undergoes an ergodicity breaking transition at $p_G$.

In this work we are interested in studying the properties of out-of-equilibrium algorithms and how they compare with the thermodynamic picture detailed in [15]. Before ending this section it is useful to mention [15] that the model can be generalized by considering a positive definite function $G(z)$ and defining the gaps as Gaussian random functions with statistics

$$\overline{h_\mu(\underline{x})} = 0 \qquad \overline{h_\mu(\underline{x}) h_\nu(\underline{y})} = \delta_{\mu\nu} G\left(\frac{\underline{x} \cdot \underline{y}}{N}\right) \,. \tag{5}$$

Here the overline stands for the average over the realization of the functions $h_\mu$: in the practical case in Eq. (2) this average is nothing but the average over the realization of the random matrices $J^\mu$. A practical way to realize Eq. (5) is by generalizing the $h_\mu$s to be a sum of terms each of which is a random tensor contraction with the vector $\underline{x}$, see [15] for more details. In the particular case of Eq. (2) we have that $G(z) = z^2/2$ and, in the following, we will always consider this case when a specification of $G(z)$ is required. Finally we note that if $G(z)$ is linear, the model itself becomes linear (apart from the spherical constraint imposed on $\underline{x}$) and there is no replica-symmetry-breaking, see [15] and [28] for a related model.

## 2.2 The dynamics

We consider the following gradient-based dynamical equations

$$\dot{x}_i(t) = -\mu(t) x_i(t) + \frac{\partial H}{\partial x_i} + \eta_i(t) \,, \tag{6}$$

where the Lagrange multiplier $\mu(t)$ is self-consistently determined in order to enforce that $|\underline{x}(t)|^2 = N$ at all times. The initial condition of Eq. (6) is chosen at random uniformly over the entire phase space. The noise $\eta(t)$ can be thought as being Gaussian with zero mean and two-point function given by

$$\langle \eta_i(t)\eta_j(t') \rangle = \delta_{ij}\Gamma(t,t'), \tag{7}$$

If $\Gamma(t,t') = 2T\delta(t-t')$ one has a Langevin dynamics at temperature $T$. Unless there is replica symmetry breaking, such dynamics ends up on a stationary state provided by the Gibbs measure at the corresponding temperature. If the kernel $\Gamma(t,t')$ is time translational invariant and characterized by a *persistent* time $\tau$[1] one ends up with a model which can be thought as describing the effect of active noise characterized by a microscopic timescale $\tau$. An exponentially decaying kernel $\Gamma$ would then correspond to active Ornstein-Uhlenbeck dynamics [23]. Here we will leave free the form of $\Gamma$ and write the dynamical equations for a generic kernel. Finally, if we send $\Gamma \to 0$ we recover gradient descent dynamics which we will be our main focus.

## 3 Dynamical mean field theory

We are interested in analyzing the set of differential equations in Eq. (6) in the $N \to \infty$ limit. Since the model is mean field in nature, it is possible to derive a set of closed integro-differential equations describing the dynamical correlation function in the thermodynamic limit. This can be done by using the dynamical mean field theory (DMFT). To derive the DMFT equations there are several routes. One can either use a path integral formalism following the Martin-Siggia-Rose-Jenssen-De Dominicis approach [29–32] or one can do a dynamical cavity method derivation [22, 23]. In this work we use a path integral formalism combined with the use of a Fermionic (Grassmann) algebra to simplify the formalism. This technical shortcut bears also some physical advantage. When the dynamics starts at equilibrium and stays at equilibrium, Fermionic fields can be transformed into Bosonic ones leaving unaltered the form of the action of the path integral. The corresponding supersymmetry (SUSY) encodes the fluctuation-dissipation theorem [26]. Therefore this approach is typically called a SUSY derivation of the DMFT equations.

The starting point is to write down the dynamical partition function which, by causality of the dynamics, is always equal to one:

$$Z_{\text{dyn}} = 1 = \left\langle \int \mathcal{D}\underline{x}(t)\mathcal{D}\underline{\hat{x}}(t) \exp\left( i \int dt\, \underline{\hat{x}}(t) \cdot \left( -\underline{\dot{x}}(t) - \mu(t)\underline{x}(t) - \frac{\partial H}{\partial \underline{x}(t)} + \underline{\eta}(t) \right) \right) \right\rangle, \tag{8}$$

and the brackets denote the average over the realization of the noise and the initial condition of the dynamics. Since this equality holds for all realizations of the disorder we can average Eq. (8) to get

$$Z_{\text{dyn}} = \overline{\left\langle \int \mathcal{D}\underline{x}(t)\mathcal{D}\underline{\hat{x}}(t) \exp\left( i \int dt\, \underline{\hat{x}}(t) \cdot \left( -\underline{\dot{x}}(t) - \mu(t)\underline{x}(t) - \frac{\partial H}{\partial \underline{x}(t)} + \underline{\eta}(t) \right) \right) \right\rangle}. \tag{9}$$

The SUSY derivation starts by introducing a simple way to rewrite the action of the dynamical partition function. One denotes by $\theta_a$ a Grassmann variable [33] and defines

$$\underline{x}(a) = \underline{x}(t_a) + i\theta_a\underline{\hat{x}}(t_a), \qquad a = (t_a, \theta_a). \tag{10}$$

---

[1] A typical example would be to take $\Gamma(t,t') = \exp[-(t-t')/\tau]$ but scale free correlations can be also considered.

Using the algebra of Grassmann integration and averaging over the noise $\underline{\eta}$ one gets

$$Z_{\text{dyn}} = \overline{\int \mathcal{D}\underline{x}(a) \exp\left(-\frac{1}{2}\int \mathrm{d}a\,\mathrm{d}b\,\underline{x}(a)\mathcal{K}(a,b)\underline{x}(b) - \sum_{\mu=1}^{M}\int \mathrm{d}a\,v(h_\mu(a))\right)}, \quad (11)$$

where the kinetic kernel $\mathcal{K}(a,b)$ is implicitly defined as

$$-\frac{1}{2}\int \mathrm{d}a\,\mathrm{d}b\,\underline{x}(a)\mathcal{K}(a,b)\underline{x}(b) = -\int \mathrm{d}t\,i\underline{\hat{x}}(t)\cdot(\dot{\underline{x}} + \mu(t)\underline{x}(t)) - \frac{1}{2}\int \mathrm{d}t \int \mathrm{d}t'\underline{x}(t)\cdot\underline{x}(t')\Gamma(t,t'), \quad (12)$$

furthermore we have denoted by $v(h) = (h-p_0)^2/2$ and by $h_\mu(a)$ a shorthand notation for $h_\mu(\underline{x}(a))$. The integration measure $\mathrm{d}a$ is defined as $\mathrm{d}a = \mathrm{d}t\,\mathrm{d}\theta_a$. Integrating over the disorder, one can rewrite the dynamical partition function as

$$Z_{\text{dyn}} = \int \mathcal{D}\underline{x}(a) \exp\left(-\frac{1}{2}\int \mathrm{d}a\,\mathrm{d}b\,\underline{x}(a)\mathcal{K}(a,b)\underline{x}(b) + \alpha N \ln \mathcal{Z}\right), \quad (13)$$

where the local partition function $\mathcal{Z}$ can be written as

$$\mathcal{Z}\left[\frac{\underline{x}(a)\cdot\underline{x}(b)}{N}\right] = \int \mathcal{D}h(c)\,\mathcal{D}\hat{h}(c)\,e^{\mathcal{S}},$$
$$\mathcal{S} = i\int \mathrm{d}a\,h(a)\hat{h}(a) - \int \mathrm{d}a\,v[h(a)] - \frac{1}{2}\int \mathrm{d}a\,\mathrm{d}b\,\hat{h}(a)G\left[\frac{\underline{x}(a)\cdot\underline{x}(b)}{N}\right]\hat{h}(b). \quad (14)$$

It is clear that the local partition function $\mathcal{Z}$ is a function of $\underline{x}(a)\cdot\underline{x}(b)/N$. Therefore it is useful now to change integration variables, from $\underline{x}(a)$ to the dynamical overlap matrix

$$Q(a,b) = \frac{1}{N}\underline{x}(a)\cdot\underline{x}(b). \quad (15)$$

Including the Jacobian of the change of variables one gets that the dynamical partition function can be rewritten as

$$Z_{\text{dyn}} \propto \int \mathcal{D}Q(a,b) \exp\left[N\mathcal{A}[Q]\right],$$
$$\mathcal{A} = -\frac{1}{2}\int \mathrm{d}a\,\mathrm{d}b\,\mathcal{K}(a,b)Q(a,b) + \frac{1}{2}\ln\det(Q) + \alpha \ln \mathcal{Z}(Q), \quad (16)$$

and we have neglected irrelevant constant terms. For large $N$ there is a large deviation principle controlling $Z_{\text{dyn}}$. In a typical setting, see [23,25], one extracts from the local partition function an effective self-consistent non-Markovian stochastic process which can be numerically solved. In the present case, the self-consistent stochastic process would be linear and therefore integrable. Indeed the local partition function in Eq. (14) is Gaussian and therefore the integral can be done analytically. This is the main point of departure of the present work from previous works. For example, the DMFT for the, closely related, perceptron model [23] is almost identical to the present model the only difference being that in that case $v(h) = (h-p_0)^2\theta(p_0-h)/2$ and therefore the corresponding local partition function cannot be integrated explicitly since one does not get a Gaussian integral. We obtain

$$\mathcal{Z} \propto \det(G)^{-\frac{1}{2}} \int \mathcal{D}h(a) \exp\left(-\frac{1}{2}\int \mathrm{d}a\,\mathrm{d}b\,h(a)\left[G^{-1}(a,b) + \delta(b-a)\right]h(b) + p_0\int \mathrm{d}a\,h(a)\right)$$

$$\propto \det(G)^{-\frac{1}{2}}\det\left(G^{-1}+I\right)^{-\frac{1}{2}} \exp\left(\frac{p_0^2}{2}\int \mathrm{d}a\,\mathrm{d}b\,\left[G^{-1}+I\right]^{-1}(a,b)\right)$$

$$\propto \det(I+G)^{-\frac{1}{2}} \exp\left(\frac{p_0^2}{2}\int \mathrm{d}a\,\mathrm{d}b\,\left[G^{-1}+I\right]^{-1}(a,b)\right), \quad (17)$$

and we have denoted by $I$ the identity operator, namely $I(a,b) = \delta(a-b)$. Note that we have used the shorthand notation $G(a,b) \equiv G(Q(a,b))$ and that $G^{-1}$ is to be understood as the inverse operator of the kernel $G$. In the large $N$ limit, the dynamical partition function is dominated by a saddle point which can be obtained by taking the variational equation of $\mathcal{A}$ with respect to $Q$. We get

$$-\frac{1}{2}\mathcal{K}(a,b) + \frac{1}{2}Q^{-1}(a,b) + \Sigma(a,b) = 0,$$
$$\Sigma(a,b) = \alpha \frac{\delta \ln \mathcal{Z}}{\delta Q(a,b)}, \tag{18}$$

The term $\Sigma(a,b)$ is nothing but the self-energy appearing in a Dyson equation fixing the dynamical propagator $Q(a,b)$. Its strength is proportional to $\alpha$ which plays the role of a coupling constant since it also tunes the strength of the interaction between different degrees of freedom. Performing explicitly the derivatives of $\mathcal{Z}$ with respect to $Q$ we get

$$-\int \mathrm{d}c\, \mathcal{K}(a,c)Q(c,b) + \delta(a-b) - \alpha \int \mathrm{d}c\, (I+G)^{-1}(a,c)\, G'(Q(a,c))Q(c,b) +$$
$$+\alpha p_0^2 \int \mathrm{d}c\, \mathrm{d}x\, \mathrm{d}y\, G'(Q(a,c))(I+G)^{-1}(x,a)(I+G)^{-1}(y,c)Q(c,b) = 0. \tag{19}$$

In order to solve this equation we need to unfold its Grassmann structure and project it back from the to the real time space. To do that we go back to the original definition of $Q(a,b)$ in Eq. (15) we have that

$$Q(a,b) = C(t_a, t_b) + \theta_a R(t_b, t_a) + \theta_b R(t_a, t_b), \tag{20}$$

where correlation function $C(t,t')$ is defined as

$$C(t,t') = \frac{1}{N}\underline{x}(t) \cdot \underline{x}(t'). \tag{21}$$

The response function $R(t,t')$ is instead given by

$$R(t,t') = \frac{1}{N}\sum_i^N \frac{\delta x_i(t)}{\delta \eta_i(t')}, \tag{22}$$

and therefore it encodes the change in the trajectory of the system induced by a small linear kick on the rhs of the dynamical equation (6). Causality implies that $R(t \leq t', t') = 0$ and that $Q$ does not contain any additional term proportional to $\theta_a \theta_b$. The structure of $Q$ implies the following structure for the terms appearing in the self-energy

$$A(a,b) = (I+G)^{-1}(a,b) = C^A(t_a, t_b) + \theta_a R^A(t_b, t_a) + \theta_b R^A(t_a, t_b). \tag{23}$$

Indeed $G$ does not contain any term proportional to $\theta_a \theta_b$ because $R(t_a, t_b)R(t_b, t_a) = 0$ for all $t_a, t_b$ and this implies that the same is true for the kernel $A$ too. Note that Eq. (23) provides the implicit definition of the correlators $C^A$ and $R^A$ and enter in the the *self-energy* structure of

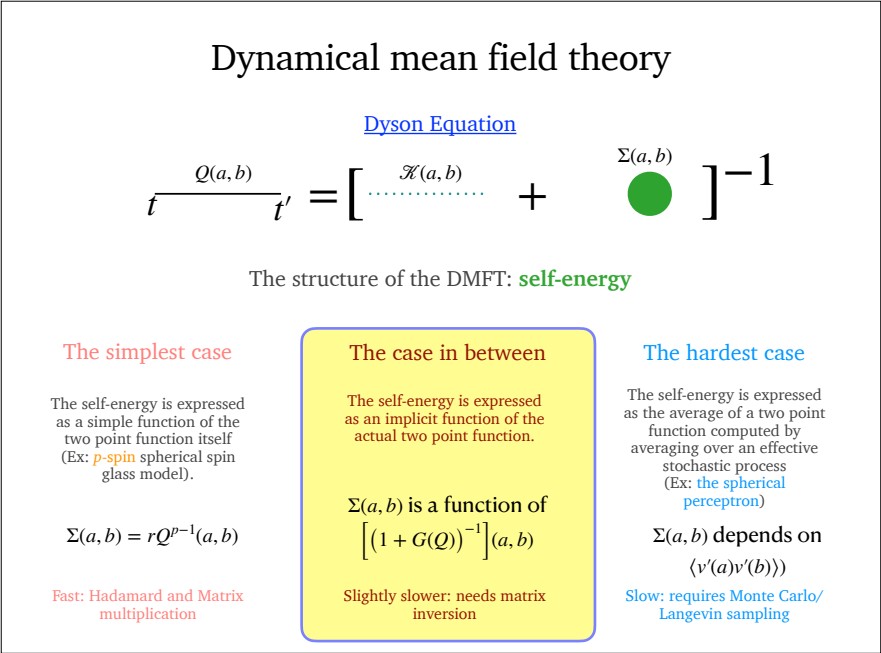

Figure 1: A comparative view of the structure of the DMFT equations. The Dyson equation is represented by Eq. (18). Different models differ by the form of the self-energy $\Sigma(a, b)$. In the spherical $p$-spin model [26] this is a simple function of the propagator itself $Q(a, b)$. In the hardest situation such as the spherical perceptron model [23], $\Sigma(a, b)$ is expressed in terms of averages over some self consistent stochastic process whose statistics depends on $\Sigma$ itself. The present case represents a situation in between the previous two where $\Sigma$ is given explicitly in terms of $Q(a, b)$ albeit it involves a complex matrix inversion.

the Dyson equation (18). Unfolding the supersymmetric algebra we get

$$
\begin{aligned}
\partial_t C(t, t') = & -\mu(t)C(t, t') + \int_0^{t'} ds\, \Gamma(t, s)R(t', s) - \alpha \int_0^{t'} ds\, C^A(t, s)G'(C(t, s))R(t', s) \\
& -\alpha \int_0^t ds\, [C^A(t, s)G''(C(t, s))R(t, s)C(s, t') + R^A(t, s)G'(C(t, s))C(s, t')] + \\
& + \alpha p_0^2 \int_0^{t'} ds \int_0^t dt_x \int_0^s dt_y\, R^A(t, t_x)R^A(s, t_y)G'(C(t, s))R(t', s) \\
& + \alpha p_0^2 \int_0^t ds \int_0^t dt_x \int_0^s dt_y\, R^A(t, t_x)R^A(s, t_y)G''(C(t, s))R(t, s)C(s, t'), \\
\partial_t R(t, t') = & -\mu(t)R(t, t') + \delta(t - t') + \\
& -\alpha \int_{t'}^t ds\, [C^A(t, s)G''(C(t, s))R(t, s)R(s, t') + R^A(t, s)G'(C(t, s))R(s, t')] + \\
& + \alpha p_0^2 \int_{t'}^t ds \int_0^t dt_x \int_0^s dt_y\, R^A(t, t_x)R^A(s, t_y)G''(C(t, s))R(t, s)R(s, t'),
\end{aligned}
\tag{24}
$$

Note that in deriving these equations we have used that $R^A(t, s > t) = 0$ which it can be shown it is a consistent solution of the equations for $R^A$.

The equation for the Lagrange multiplier $\mu$ can be obtained by imposing that $dC(t,t)/dt = 0$ which gives

$$
\begin{aligned}
\mu(t) = &-\alpha \int_0^t ds [C^A(t,s)G''(C(t,s))R(t,s)C(s,t) + R^A(t,s)G'(C(t,s))C(s,t)] \\
&+ \alpha p_0^2 \int_0^t ds \int_0^t dt_x \int_0^s dt_y R^A(t,t_x)R^A(s,t_y)G''(C(t,s))R(t,s)C(s,t) \\
&+ \alpha p_0^2 \int_0^t ds \int_0^t dt_x \int_0^s dt_y R^A(t,t_x)R^A(s,t_y)G'(C(t,s))R(t,s) \\
&- \alpha \int_0^t ds\, C^A(t,s)G'(C(t,s))R(t,s) + \int_0^t ds\, \Gamma(t,s)R(t,s)\,.
\end{aligned}
\tag{25}
$$

The last step to close the dynamics is to provide an equation for $C^A$ and $R^A$. Unfolding the definition of $A$ in terms of $Q$ we get the following equation

$$
\int dt_c \mathcal{M}(t_b,t_c) \begin{pmatrix} C^A(t_a,t_c) \\ R^A(t_a,t_c) \end{pmatrix} = \begin{pmatrix} 0 \\ \delta(t_a - t_b) \end{pmatrix},
\tag{26}
$$

where the matrix $\mathcal{M}(t_c,t_b)$ is given by

$$
\mathcal{M}(t_b,t_c) = \begin{pmatrix} \delta(t_c - t_b) + G'(C(t_c,t_b))R(t_b,t_c) & G(C(t_c,t_b)) \\ G''(C(t_b,t_c))R(t_b,t_c)R(t_c,t_b) & \delta(t_c - t_b) + G'(C(t_c,t_b))R(t_c,t_b) \end{pmatrix}.
\tag{27}
$$

This tells us that the way to get $C^A$ and $R^A$ is by computing the inverse of $\mathcal{M}$ and applying it to the rhs of Eq. (26). Note that causality implies that $R(t_b,t_c)R(t_c,t_b) = 0$ and therefore the matrix to be inverted has an upper triangular shape which is useful for computational purposes. Since the computation of $C^A$ and $R^A$ is the bottleneck for the computation of the self-energy of the Dyson equation, we clearly see here that the present model has some technical advantages. A comparison between different structures of the self-energy of the Dyson equation in DMFT is summarized in Fig. 1.

## 4 Numerical integration scheme of the DMFT equations

In this section we provide a numerical scheme to solve the DMFT equations. We will discretize the time in small time intervals. The discretized dynamics cannot be tracked exactly because of the spherical constraint. Indeed the Lagrange multiplier $\mu$ accounts for the spherical constraint on the vector $\underline{x}$ only to leading order in the time interval and therefore it is exact only in the continuous time limit. However we can always solve the equations in an approximate way by performing a small timestep discretization. The purpose of this section is to detail the numerical procedure to do this and to compare and validate the results with the numerical simulations of finite size systems. We will assume that the DMFT equations are discretized with a timestep $dt$ and that time is an integer multiple of $dt$. Therefore we will use the shorthand notation $C_{i,j} \equiv C(t = idt, t' = jdt)$ and two point functions become matrices while one point functions become vectors.

### 4.1 Correlation and Response function

We consider first Eqs. (24). A straightforward discretization of the equations based on the Euler scheme gives

$$
\begin{aligned}
C_{i+1,j} - C_{i,j} = \mathrm{d}t &\left[ \mathrm{d}t \sum_{k=0}^{j} \Gamma_{ik} R_{j,k} - \alpha \left( \mathrm{d}t \sum_{k=0}^{i} \left( C_{i,k}^{A} G'(C_{i,k}) R_{j,k} + C_{i,k}^{A} G''(C_{i,k}) R_{i,k} C_{k,j} \right. \right. \right.\\
&\left. + R_{i,k}^{A} G'(C_{i,k}) C_{k,j} - p_0^2 \mathrm{d}t^2 \left( \sum_{l=0}^{i} R_{i,l}^{A} \right) \left( \sum_{l=0}^{k} R_{k,l}^{A} \right) (G'(C_{i,k}) R_{j,k} + G''(C_{i,k}) R_{i,k} C_{k,j}) \right) \right) - \mu_i C_{i,j} \right]
\end{aligned}
$$

$$
\begin{aligned}
R_{i+1,j} - R_{i,j} = \delta_{ij} + \mathrm{d}t &\left[ -\mu_i R_{i,j} - \alpha \left( \mathrm{d}t \sum_{k=j}^{i} \left( C_{i,k}^{A} G''(C_{i,k}) R_{i,k} R_{k,j} + R_{i,k}^{A} G'(C_{i,k}) R_{k,j} \right. \right. \right.\\
&\left.\left.\left. - p_0^2 \mathrm{d}t^2 \left( \sum_{l=0}^{i} R_{i,l}^{A} \right) \left( \sum_{l=0}^{k} R_{k,l}^{A} \right) R_{i,k} R_{k,j} G''(C_{i,k}) \right) \right) \right].
\end{aligned}
\tag{28}
$$

Note that we have used the fact that $R_{i,j} = 0$ if $i \leq j$ and furthermore that $C_{i,j} = C_{j,i}$. The spherical constraint is imposed by fixing $C_{i,i} = 1$ and by propagating the Lagrange multiplier $\mu$ as

$$
\begin{aligned}
\mu_i = \mathrm{d}t \sum_{k=0}^{i} \Gamma_{i,k} R_{i,k} - \alpha &\left[ \mathrm{d}t \sum_{k=0}^{i} \left( C_{i,k}^{A} G'(C_{i,k}) R_{i,k} + C_{i,k}^{A} G''(C_{i,k}) R_{i,k} C_{k,i} + R_{i,k}^{A} G'(C_{i,k}) C_{k,i} \right. \right.\\
&\left.\left. - p_0^2 \mathrm{d}t^2 \left( \sum_{l=0}^{i} R_{i,l}^{A} \right) \left( \sum_{l=0}^{k} R_{k,l}^{A} \right) (G'(C_{i,k}) R_{i,k} + G''(C_{i,k}) R_{i,k} C_{k,i}) \right) \right].
\end{aligned}
\tag{29}
$$

These equations have a causal structure and can be easily integrated. They depend on $C^A$ and $R^A$ and therefore we need to provide an equation for them.

### 4.2 The self-energy

Both $R^A$ and $C^A$ enter in the structure of the self-energy of the Dyson equation (18). They are given as the solution of Eq. (26). We now need to solve this equation properly discretized in time. It is important to note that we need two quantities $C^A$ and $R^A$ that have a matrix structure because they are functions of two time indices. Accordingly, the matrix defined on the lhs of the Eq. (26) has a tensorial structure and therefore we need to parametrize it in a way that one can use linear algebra to perform the inverse of the kernel operator $\mathcal{M}$ entering in the equation. In order to do that, we transform the linear system in Eq. (26) into the form

$$
\sum_{j=0}^{2L-1} \Lambda_{ij} \nu_j = w_i(t_a),
\tag{30}
$$

and we have denoted by $L$ the total time interval of the numerical integration. Assuming that $t_a = a\mathrm{d}t$ and $t_b = b\mathrm{d}t$ for $a$ and $b$ integers, we can write the the vectors $\underline{v}$ and $\underline{w}$ with the following encoding:

$$
\begin{aligned}
w_i(t_a) &= \begin{cases} 0 & i < L \\ \dfrac{\delta_{a,i}}{\mathrm{d}t} & i \geq L \end{cases} \\
\nu_j &= \begin{cases} C_{aj}^{A} & j < L \\ R_{aj}^{A} & j \geq L. \end{cases}
\end{aligned}
\tag{31}
$$

Finally the matrix $\Lambda$ assumes the following form

$$\Lambda_{ij} = \begin{cases} \delta_{ij} + \mathrm{d}t\, G'(C_{ij}) R_{ji} & i,j < L \\ \delta_{ij} + \mathrm{d}t\, G'(C_{ij}) R_{ij} & i,j \geq L \\ 0 & i \geq L,\ j < L \\ \mathrm{d}t\, G(C_{ij}) & i < L,\ j \geq L\,. \end{cases} \tag{32}$$

Therefore, for each value of $a$ one can invert the linear system and solve for $C^A$ and $R^A$. Collecting different values of $a$ one gets the full matrices $C^A$ and $R^A$.[2] Note that since the matrix $\Lambda$ is an upper triangular block matrix, one can invert it recursively and therefore the code to implement the computation of $C^A$ and $R^A$ can be made more efficient.

## 5  Results on Gradient Descent dynamics

In this section we report the results of the numerical integration of the DMFT equations. We will first validate the equations and their integration with a comparison with numerical simulations. Then we will discuss the behavior of the DMFT solution across the phase diagram of the model. In this section we will focus on gradient descent dynamics which means that we consider $\Gamma = 0$.

### 5.1  Comparison with numerical simulations

In order to validate the DMFT equations and their numerical integration we first compare them with the results of the numerical simulations of the model. The algorithm we use to perform the numerical simulations goes as follows. We initialize the configuration of the system at random with a flat measure on the sphere[3] $|\underline{x}(0)|^2 = N$. Then we discretize time in small timesteps of size $\mathrm{d}t$ and we update the configuration of the system according to

$$\underline{x}(t+\mathrm{d}t) = \sqrt{N}\, \frac{\underline{x}(t) - \mathrm{d}t\, \frac{\partial H}{\partial \underline{x}}}{|\underline{x}(t) - \mathrm{d}t\, \frac{\partial H}{\partial \underline{x}}|}, \tag{33}$$

which is nothing but a discretized gradient descent step followed by a projection on the sphere $|\underline{x}|^2 = N$. We run the simulations for $N = 500$ and $M = 125$ and we consider a time interval $\mathrm{d}t = 0.025$. For each value of $p_0$ we run 1000 simulations. Each sample is identified by the initial condition of the dynamics and the realization of the matrices $J^\mu$ as well. Changing sample means that we change both initial conditions and the disorder. For each sample we measure a series of dynamical quantities which are then averaged over the different samples. The quantities we compute are the following:

- We measure the average value of $h_\mu$, namely

$$\langle h(t) \rangle = \frac{1}{M} \sum_{\mu=1}^{M} h_\mu(t)\,. \tag{34}$$

  It is also useful to consider $|\langle h(t) \rangle - p_0|$ since this function goes to zero when the dynamics reaches a zero energy configuration and it goes to a finite positive value when the

---

[2]The matrix $C^A$ is symmetric in its arguments as it can be checked directly from the equations.
[3]Practically we extract $y_i$ from $\mathcal{N}(0,1)$ and fix $x_i = y_i \sqrt{N}/|\underline{y}|$.

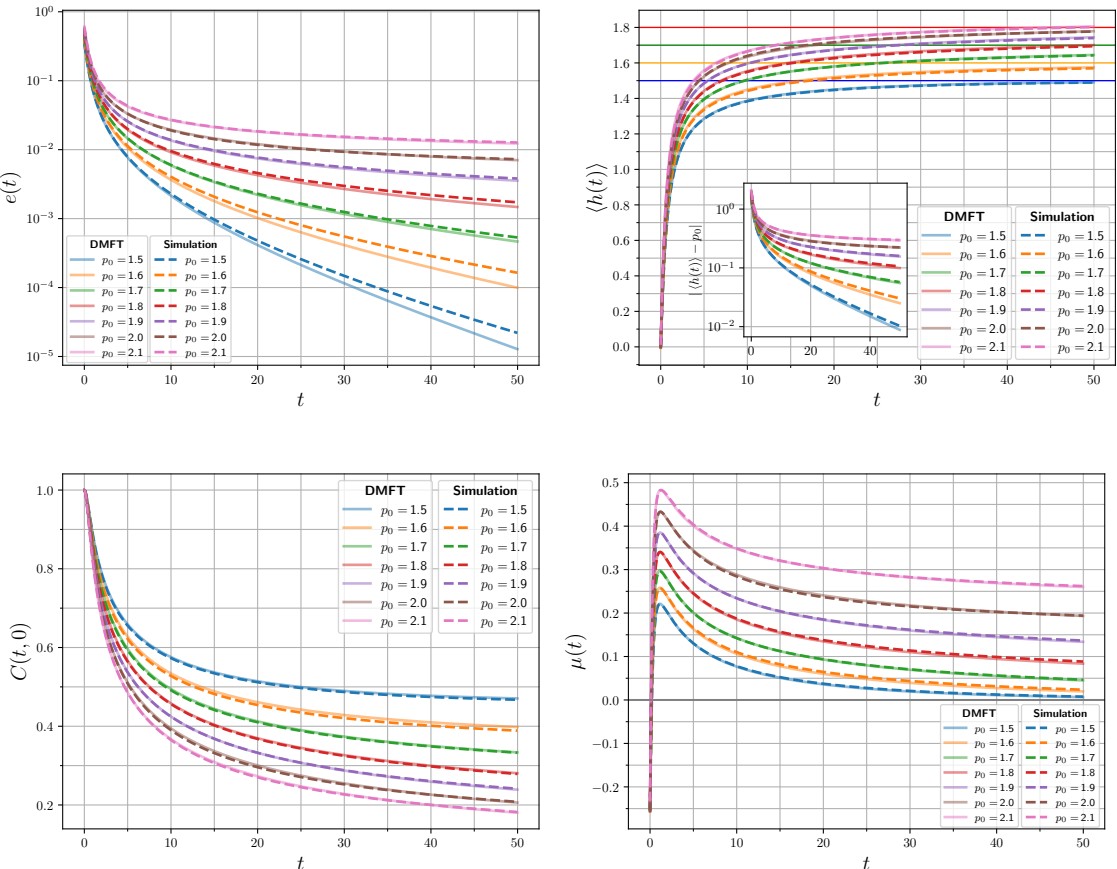

Figure 2: Comparison between numerical simulations and the numerical integration of the DMFT equations.

dynamics lands on a local minimum of the Hamiltonian at positive energy. The DMFT expression for $\langle h(t) \rangle$ is given by

$$\langle h(t) \rangle = \langle h(a) \rangle_{\mathcal{Z}}|_{\theta_a=0} \,, \tag{35}$$

and we have indicated on the right hand side the average with respect to the measure identified by the partition function $\mathcal{Z}$ defined in Eq. (14). It is easy to show that in order to compute $\mathcal{Z}$ one can promote $p_0$ to take a Grassmann variable dependence $p_0 \to p_0(a)$ and that

$$\langle h(a) \rangle = \frac{\delta}{\delta p_0(a)} \ln \mathcal{Z}\bigg|_{p_0(a) \to p_0} \,, \tag{36}$$

from which we get that

$$p_0 - \langle h(t) \rangle = p_0 \int_0^t \mathrm{d}s R^A(t,s), \tag{37}$$

which provides a direct physical interpretation of $R^A(t,s)$.

• We measure the energy per degree of freedom as a function of time defined as

$$e(t) = \frac{1}{N} H(t). \tag{38}$$

Using the same strategy that we employed to compute $\langle h(t) \rangle$, we can show that the DMFT expression for the energy is simply given by

$$e(t) = \frac{\alpha}{2} \left[ p_0^2 \left( \int_0^t \mathrm{d}s R^A(t,s) \right)^2 - C^A(t,t) \right]. \tag{39}$$

The correlation function $C^A(t,s)$ admits the simple interpretation given by

$$\langle h(t)h(s) \rangle - \langle h(t) \rangle \langle h(s) \rangle = \frac{1}{M} \sum_{\mu=1}^{M} h_\mu(t) h_\mu(s) - \frac{1}{M^2} \sum_{\mu\nu=1}^{M} h_\mu(t) h_\nu(s) = -C^A(t,s). \tag{40}$$

- We measure the correlation function of the system between its initial configuration and the configuration at time $t$:

$$C(t,0) = \frac{1}{N} \underline{x}(t) \cdot \underline{x}(0). \tag{41}$$

- Finally, we measure the Lagrange multiplier $\mu(t)$. It is possible to show that its microscopic expression is given by

$$\mu(t) = -\frac{1}{N} \sum_{\mu=1}^{M} (h_\mu(t) - p_0) \sum_{i=1}^{N} \frac{\partial h_\mu(t)}{\partial x_i(t)} x_i(t), \tag{42}$$

and its DMFT expression is given by Eq. (25).

In Fig.2 we plot these quantities measured from simulations and as found via the numerical integration of the DMFT equations and we observe a rather good agreement. We also see that when looking at quantities that go to zero at large time in logarithmic scale, such as the energy or $|\langle h(t) \rangle - p_0|$, the agreement becomes less good below $10^{-3}$. This should be also expected since while simulations are on finite size and finite number of samples, we also know that the agreement with the DMFT cannot be perfect due to the fact that we are integrating the dynamics, both in numerical simulations and DMFT, at finite timestep $\mathrm{d}t$ while we should expect perfect agreement only for $\mathrm{d}t \to 0$. In order to establish the dependence of the precision of the numerical integration of the DMFT equations from the timestep $\mathrm{d}t$ we follow [34] and consider

$$\Delta C(t) = \frac{|C_{\mathrm{d}t}(t,0) - C_{\mathrm{d}t=0.025}(t,0)|}{C_{\mathrm{d}t}(t,0)}, \tag{43}$$

where $C_{\mathrm{d}t}(t,0)$ is the correlation function $C(t,0)$ obtained from the DMFT equations integrated with timestep $\mathrm{d}t$. We plot $\Delta C(t)$ in Fig.3 for four different values of $t$. We observe that the numerical integration error is quite small and appears to be approximately linear in $\mathrm{d}t$. We remark also that the slope of the curves is a decreasing function of $t$. This means that the numerical integration error is smaller at large times and this is probably due to the fact that the DMFT equations involve sums which average out as they extend to larger and larger times.

## 5.2 Gradient descent dynamics beyond the zero temperature replica symmetry breaking transition

In this section we report the results of the numerical integration of the DMFT equations to explore the performances of gradient descent dynamics for different values of $p_0$ which is the relevant control parameter of the model. In Fig.4 we plot the energy and $\langle h(t) \rangle$ as a function

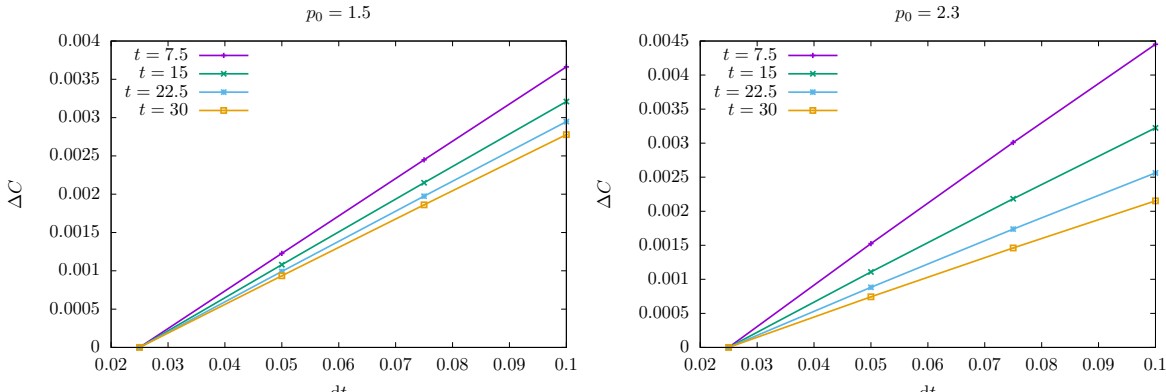

Figure 3: The dependence of the correlation function $C(t,0)$ on the integration timestep $dt$. We plot $\Delta C(t)$ as defined in Eq. (43) for four different values of $t$. The right panel corresponds to the integration for $p_0 = 2.3$ while the left panel we plot the same quantities for $p_0 = 1.5$.

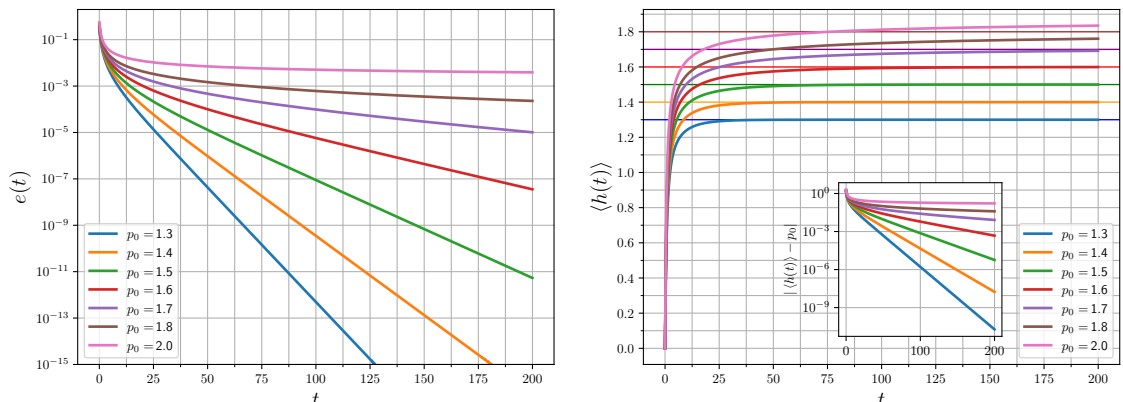

Figure 4: Numerical integration of the DMFT equation on long timescales. The energy and $\langle h(t) \rangle$ as a function of time. It is clear that for $p_0 \leq 1.7$ both quantities suggest an exponential decay to zero.

of time for different values of $p_0$ at $\alpha = 1/4$. The energy decays exponentially to zero for $p_0$ small enough while $\langle h(t) \rangle$ converges to $p_0$. We first note that gradient descent finds zero energy configurations well beyond the replica symmetry breaking transition of the zero temperature Gibbs measure which is at $p_0 = 1$ for $\alpha = 1/4$, see [15]. This implies that RSB transition in the Gibbs measure at zero temperature may be totally irrelevant for the effectiveness of gradient descent dynamics in finding zero energy configurations.[4] This is not in contradiction with RSB theory: the zero temperature RSB transition signals that the space of solutions of the CCSP cannot be explored ergodically by a Langevin dynamics in the zero temperature limit on short timescales. However since gradient descent finds configurations at zero energy on a short timescale (meaning exponentially fast) it essentially stops. This means that the stationary measure of gradient descent may be different from the zero temperature limit of the Gibbs measure and therefore even if the latter develops a complex structure of pure states, this does not automatically mean that gradient descent fails in finding zero energy configurations.

---

[4]A case where RSB at zero temperature is relevant to understand the success of Greedy Algorithms to solve random CSPs is the one of locked (discrete) constraint satisfaction problems (CSP) [35].

Moreover in the context of the spherical perceptron problem, numerical simulations [36] seem to indicate that that Gradient Descent finds solutions also after the RSB transition of the flat measure on the solution space (the zero temperature limit of the Gibbs measure). Our results clearly confirm these findings in the context of the model considered in this manuscript.

Looking at Fig.4, one would conjecture also that the gradient descent satisfiability transition $p_J^{GD}$ is located in the range $1.8 < p_J^{GD} < 2$ which must be compared with the thermodynamic satisfiability transition point located at $p_J \simeq 1.87$ estimated numerically in [15]. In order to estimate better $p_J^{GD}$ we have considered the following procedure. Given that the energy in the SAT phase decreases exponentially fast, we have measured the relaxation time $\tau(p_0)$ as the time it takes for the dynamics to bring the system to an energy below a threshold value. We fix this value to be $e_{th} = 10^{-6}$. In the left panel of Fig.5 we plot $\tau(p_0)$ at $\alpha = 1/4$. We plot data from two different datasets obtained by changing the integration timestep $dt$ to confirm that our estimate of $\tau$ is weakly dependent on this integration parameter. We observe that the relaxation time seems to have a divergence when increasing $p_0$. Therefore we can plot $\tau$ as a function of $p_J^{GD} - p_0$ in a logarithmic scale, and assuming that $\tau$ has a power law divergence, we make a rough estimate of $p_J^{GD} \simeq 1.86$. In the right panel of Fig.5 we plot the result of this analysis. The estimated value of $p_J^{GD}$ is very close to the thermodynamic SAT/UNSAT transition point. Whether the two transitions are actually the same remains unclear and one needs to refine the numerical estimation and construct a theory for the transition points too. If the two transitions are the same, there should be a way to show that the DMFT equations reduce to the RSB equations derived in [15, 37] exactly at the transition. A careful analysis of this point is left for future work. However, it could happen that while the location of the satisfiability point is algorithm dependent see the case of the packing problem [38], the critical properties of the configurations at the critical point may be universal as it happens for isostatic systems [19, 39–42]. In this case RSB is a way to understand criticality through stochastic stability, an idea that has been suggested in [43]. How these properties extend to the present model remains to be investigated.

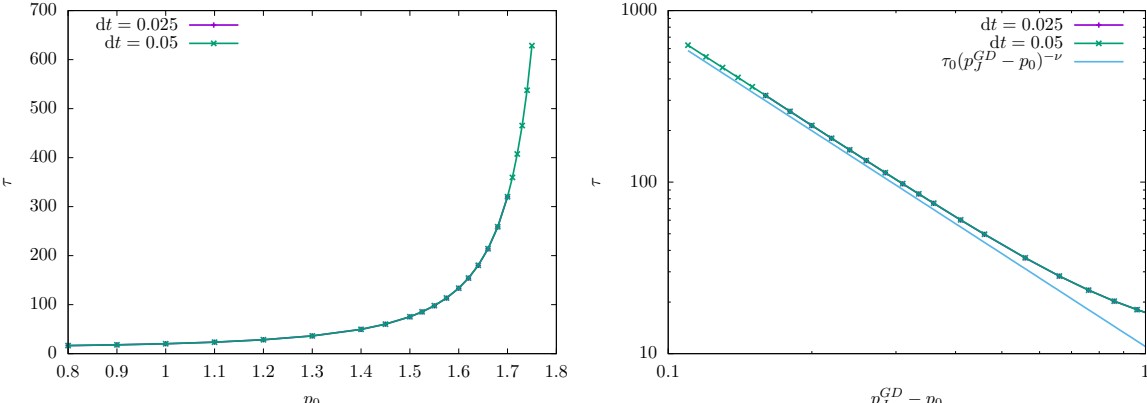

Figure 5: *Left panel:* The relaxation time as a function of $p_0$. *Right panel*: the relaxation time plotted as a function of $p_J^{GD} - p_0$ being $p_J^{GD} = 1.86$. In logarithmic scale we observe a power law divergence and we plot a tentative fit with parameters $\tau_0 = 12$ and $\nu = 1.8$.

Finally, in the left panel of Fig.6 we plot the correlation function $C(t, 0)$ as a function of time for different values of $p_0$. The correlation function goes to a positive plateau when the dynamics ends up at zero energy, sufficiently far from the algorithmic satisfiability transition. This means that starting from a random initial configuration on the sphere, the dynamics does not decorrelate completely from it and there are zero energy reachable configurations close

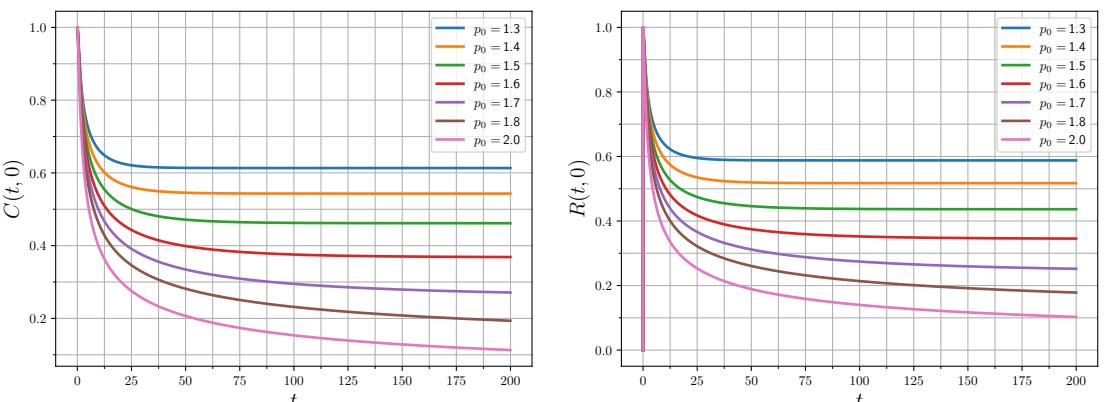

Figure 6: The correlation and response function as a function of time. For $p_0$ sufficiently small and corresponding to values where the behavior of the energy suggests an exponential decay to zero, both quantities reach a plateau at long times.

to the initial condition. Decorrelation seems more plausible as soon as $p_0$ approaches the satisfiability transition point. However it is also true that for $p_0 = 2$ which is larger than the satisfiability point, $C(t, 0)$ decays very slowly and it is unclear if it converges to zero or if it stays positive. In the right panel of Fig.6 we also plot the response function $R(t, 0)$. Interestingly this function reaches a positive plateau when $p_0$ is sufficiently small and this mirrors what is found in the context of jamming of spheres, see [44].

## 6 Perspectives

We have described the dynamical mean field theory for simple models of confluent tissues viewed as high-dimensional random continuous constraint satisfaction problems with equality constraints solved via the optimation of the square loss function. We believe that our approach opens several interesting directions and we outline them here.

- A systematic study of the DMFT equations in the regime where the noise is finite is mandatory. While for thermal noise, one can use the fact that when the dynamics equilibrates the system, this is described by a Boltzmann probability distribution, for out-of-equilibrium noise this is not true anymore and therefore it is interesting to study the rigidity transition point directly from the DMFT equations. This will be useful to address the critical properties of the rigidity transition in models of confluent tissues at the mean field level.

- A systematic study of the algorithmic SAT/UNSAT transition with the gradient descent dynamics can be done. In this work we have showed that the algorithmic satisfiability transition is close to the thermodynamic one and this was already found in finite size numerical simulations in [15]. Furthermore we have shown that the relaxation time follows a power law divergence close to the transition and we have measured the corresponding critical exponent. It would be interesting to compute this exponent explicitly and to compare it with Vertex or Voronoi models to understand the effect of the dimensionality of the system on the critical properties of the transition point.

- In deep learning applications, one typically solves the optimization problem via stochastic gradient descent (SGD) which is an algorithm where the gradient of the loss is approximately estimated by a time-changing subset (or minibatch) of the full dataset. The effectiveness of the algorithm to find zero energy configurations is a crucial point to understand deep learning. A statistical mechanics approach to address this question has been recently put forward in [25, 45, 46] where a DMFT analysis for the SGD algorithm and some interesting variations was developed. However the main problem of these works is that the form of the DMFT equations is complicated and it is difficult to extract long time results. One possibility to overcome this difficulty is to extend the DMFT analysis developed here to take into account the effect of SGD. This can be done in at least two models:

  – *Random CCSP with equality constraints.* – We can consider the model analyzed here and look at the persistent-SGD dynamics discussed in [25] and defined as

  $$\dot{x}_i(t) = -\mu(t)x_i(t) - \sum_{\mu=1}^{M} s_\mu(t)(h_\mu(t) - p_0)\frac{\partial h_\mu(t)}{\partial x_i(t)},\qquad(44)$$

  where the selection variables $s_\mu(t)$ are binary and evolve according to a Poisson process whose characteristic time $\tau$ is called persistence time. The average value of $s_\mu(t)$ is fixed to $b$ which is the (extensive) mini-batch size of the algorithm. It would be interesting to understand whether such algorithm is effective as gradient descent in finding ground states of the problem.

  – *High-dimensional inference.* – An interesting model, considered in [47] describes the following non-linear inference problem. A random vector $\underline{w}^*$ is measured according to the following set of non-linear measurements:

  $$y_\mu = \frac{1}{N}\sum_{i<j} J_{ij}^\mu w_i^* w_j^*\qquad(45)$$

  and the matrices $J^\mu$ have the same statistics of the model considered in the present work. The inference problem is to reconstruct $\underline{w}^*$ starting from the knowledge of $y_\mu$ and the matrices $J^\mu$. A way to solve the problem is by finding the vector $\underline{w}$ that minimizes the square loss Hamiltonian:

  $$H[\underline{w}] = \frac{1}{2}\sum_{\mu=1}^{M}\left(y_\mu - \sum_{i<j} J_{ij}^\mu w_i w_j\right)^2.\qquad(46)$$

  While in [47] the solution of the inference problem was analyzed from the point of view of the zero temperature limit of the posterior measure over the signal $\underline{w}$, it would be interesting to extend the DMFT analysis developed here to understand where gradient descent dynamics and SGD are able to recover the signal.

We believe that the analysis and tools presented in this work will be useful to address the questions outlined above and a detailed investigation is left for future work.

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
