# Peer review of "Dynamical mean field theory for models of confluent tissues and beyond"

_SciPost Physics, doi:SciPost Phys. 15, 219 (2023)_

## Round 2 · Referee Report · Anonymous (Referee 1) · 2023-9-14

Strengths

1 - The manuscript formulates and derive the dynamical mean-field theory of a continuous constraint-satisfaction problem used as a candidate model for the rigidity transition in confluent tissues. The dynamical equations derived are far from trivial and constitute a powerful tool to analyze the system.

2 - The manuscript also provides a numerical solution of gradient descent dynamics in fair agreement with numerical simulations

3 - The authors interpret the findings in the framework of the SAT/UNSAT transitions and argue that ergodicity breaking in Gibbs measure is irrelevant at zero temperature

Weaknesses

1 - The gradient descent dynamics is inconclusive about the long-time limit behavior when approaching the rigidity transition; a clear identification of the transition and its comparison with the thermodynamic jamming point are missing

2 - While the model is clearly related to learning and inference problems, the relationship with previous results on confluent tissues mentioned in the introduction and the contribution given by the findings sounds a bit obscure and would deserve more attention

Report

The authors analyse the dynamics of a recently introduced continuous constraint-satisfaction problem - CCSP, see Ref. [5] - as a suitable model to study the rigidity transition in confluent tissues. To this aim, starting from the microscopic dynamics defined in Eq. (6) they derive the dynamical mean-field theory (DMFT) describing the dynamics of the system in the infinite-dimensional limit via the path integral SUSY formalism.

After lengthy computations, the authors obtain dynamical equations for the correlators and the Lagrange multiplier of the system in a general framework. The mean-field nature of the system makes the dynamical equation closed without the need of self-consistent relations as in Ref. [12].

The authors finally study the zero-temperature gradient descent dynamics by simulating the microscopic dynamics of the model and solving numerically the DMFT equations. The agreement of the two methods shown in Fig. 1 is very good. The author then assume the equivalence of the two methods and focus on the DMFT solution: the exponential decay of the energy at least for 1.3 < p < 1.6 (being p the target shape, the control parameter governing the rigidity transition) show that gradient descent can find zero energy configurations even when the Gibbs measure is RSB. The convergence of the gaps to their target value is consistent with this picture. Finally, the authors show the correlation and response function computed through DMFT.

I find the paper scientifically valid, rigorous and the derivation of DMFT equations for this CCSP is surely a matter of interest for the community and the basis to explore the dynamics of such models. The SUSY derivation is compact and the authors provide a physical interpretation of the various terms, which enhances the analysis.

Furthermore, the numerical solution of DMFT is a difficult technical task that the authors solved properly. Although they have been helped in that by the mean-field nature of the equations, the authors have also managed to avoid the many pitfalls that can occur in these cases.

I think that the overall results of the paper are surely novel, rigorous and interesting to the scientific community. Nevertheless, I think that the paper does not answer some questions which they could somewhat easily tackle. In order of importance:

1) I find the results inconclusive about the SAT/UNSAT transition. All the curves presented show a clear trend with increasing p_0, but it is hard to see a qualitative change. The authors mention that the location of p_J is an ongoing task that will be published in a separate work; nonetheless I think it would be good to see it in this work, since it could give a quantitative result and a reference for further studies. A similar analysis could apply to the behavior of correlation and response functions, where the authors admit that "it is unclear if it (ed: the correlation function) converges to zero or if it stays positive." I believe that fitting the data already presented to obtain a long-time limit of the energy and a characteristic time is at hand. Although affected by possible preasymptotic behavior, it is a relevant benchmark that would considerably improve the paper.

2) While I find quite clear the relationship of the presented model with machine-learning and inference problems, the authors claim their model to be an appropriate model for the physics of confluent tissues, but this connection is somewhat lost throughout the paper. However, numerical results could be used to elaborate on the connection between the two. More specifically: - confluent cell models are typically developed at d=2,3 while the CCSP model proposed lives in a high-dimensional space. What is the impact of dimensionaity on the collective behavior and phase transitions? - Refs. [1-4] mainly refer to self-propelled vertex models, but self-propulsion is disregarded for the gradient descent case. A good reference for this topic is Bi et al., Nat. Phys. 11, 1074–1079 (2015), where the authors analyse the gradient descent of a passive 2d vertex model. In Fig. 3 of the mentioned paper the authors also show the density of states approaching the solid-fluid transition from the solid phase. Looking also at the perspective sections of Ref. [5], could the results shown include an analysis of the density of states for the proposed model (following what has been done e.g. in Ref. [29]), which could then be compared to results in vertex models? And if not, may the authors state why?

3) the irrelevance of RSB in the dynamics is most likely associated to the zero-temperature nature of gradient descent dynamics. Can the authors clarify if this behavior is general, e.g. for the case of the perceptron which also exhibits a RSB phase as studied in [8], or if it is unique to CCSPs with equalities?

4) the numerical solution of DMFT equations at very long times is notoriously a difficult task because of memory terms slowing down the computations. However, the mean-field structure of the case comes to rescue: have the authors tried to attain logarithmic time scales by means of decimation algorithms as the ones proposed by Fuchs et al 1991 J. Phys.: Condens. Matter 3 5047, and Gruber et al Phys. Rev. E 94, 042602 (2016) ? I acknowledge that changing the integration method would require a considerable work and this may be also devoted to a following paper; nonetheless I encourage the authors to try to apply it to this problems in order to get conclusive results over the long-time limit.

5) did the authors study the numerical error for finite N in simulations and finite dt in numerical solution of DMFT? Can they provide a scaling and/or an argument to assess its significance?

My overall feeling is that this is a good manuscript, the presented science is almost surely valid and it represents a step forward in the field. However at present stage the article misses some points which are the main reasons to derive the heavy DMFT formalism and that would make the paper surely worth for publication. I would recommend publication if the authors satisfyingly make all the requested changes.

Minor points and typos:

  • parenthesis missing (possible after G(Q...)) in the 6th and 7th row of Eq. (21)
  • the structures of the sentence and generally speaking the use of punctuation (misplaced commas etc.) worsen the readability of the paper
  • white space between figures could be reduced to increase the size of plots and labels therein which, even on a large screen, are still quite small to be fairly appreciated

Requested changes

1) include an analysis of the p_J location for the gradient descent dynamics based on existing data

2) comment on the relationship with vertex models, especially referring to the impact of dimensionality and the density of states

3) comment on the generality or uniqueness of the RSB irrelevance in zero-temperature dynamics

4) include a numerical analysis and/or a scaling argument for the finite-size and finite-timestep error in simulations and numerical solutions

  • validity: high
  • significance: ok
  • originality: good
  • clarity: good
  • formatting: good
  • grammar: acceptable

Author:  Pierfrancesco Urbani  on 2023-09-26  [id 4012]

(in reply to Report 1 on 2023-09-14)

We thank the reviewer for the comments, criticism and suggestions. We detail here a reply to the points raised in the report and the changes made in the revised version of the manuscript. We follow the enumeration of the report.

1) We have included in the revised version of the manuscript a figure with the relaxation time (as defined from the decay of the energy below a fixed threshold) as a function of $p_0$. The relaxation time seems to diverge with a power law behavior on approaching the algorithmic SAT/UNSAT transition. The location of the divergence is very close to the thermodynamic SAT/UNSAT transition as previously observed in numerical simulations.

2) - Concerning the impact of dimensionality on critical behavior. This is a good question but we do not have a good answer. We believe that one should consider a set of observables and critical exponents and look for them in numerical simulations and within our model. However this goes much beyond the scope of this work.
 - Concerning the density of states: this has a simple behavior. Before the transition one has a delta peak at zero energy corresponding to the flat directions along the zero energy manifold. In the UNSAT phase instead the spectrum has no delta peak at zero frequency. However the behavior of the low frequency, which is very important to control the relaxation and the elasticity of the system, is unclear from numerical simulations, see Daniel M. Sussman et al 2018 EPL 121 36001. Therefore, while investigating it within our model is certainly doable, we lack more precise data on the side of numerical simulations in finite dimensional models. Anyway this goes beyond the scope of this work which focuses on dynamics.

3) We have included a discussion on the relevance of RSB for GD dynamics in other CCSPs. For example in [36] of the revised version of the manuscript, numerical simulations have been performed in the spherical perceptron problem where it is found that gradient descent can enter the RSB phase.

4) We thank the reviewer for the suggestion. Indeed the simplest thing to do is to promote the integration timestep to be time dependent itself. The only tricky point where this complicates the analysis is in the definition of the self-energy which involves a matrix inversion. In this case one needs to be careful and add a Jacobian term in the equation for the self-energy.

5) We have provided the numerical simulations only as a validation step of the DMFT equations. Understanding the finite size effects is a formidable task because numerically one needs to perform big tensor-vector contractions and store big matrices (the $J^mu$s). A careful study of finite size fluctuations goes beyond the scope of this work. As a side remark: one of the main reasons to derive the DMFT equations is to avoid numerical simulations and to get directly the behavior of the system in the infinite system size limit! Concerning the timestep of the numerical integration, we have added Fig.3 where we plot how the correlation function at different times depends on the integration step. We have added a discussion on this point in Sec. 5.1.

---

## Round 2 · Referee Report · Anonymous (Referee 2) · 2023-9-15

Report

Dynamical mean field theory for confluent tissues models and beyond Persia Jana Kamali, Pierfrancesco Urbani Arxiv-2306.06420.pdf

In this work, the authors examine a simple yet paragmatic statistical-mechanics model, namely "for a continuous constraint satisfaction problem with equality constraints". In a previous work (ref. [5]), one of the authors had shown that this model displays a similar static phase diagram to the Voronoi/vertex model for confluent biological tissues, including its so-called ‘rigidity transition’. Here the authors aim to address instead dynamical aspects of this model, within the framework of its dynamical mean-field theory (DMFT).

More specifically, they derive in the thermodynamic limit a set of exact DMFT equations and present an efficient numerical scheme to integrate these equations. Then, focusing on the special case of gradient-descent dynamics, they validate their DMFT findings by confronting them to numerical simulations on the whole original model. They highlight one specific result, which is that the dynamics/algorithm is able to find a zero-energy configuration in the replica-symmetry-breaking phase of the static phase diagram. They conclude by giving two related models, relevant for deep learning applications, for which this program could be replicated in order to investigate stochastic-gradient-descent algorithms.

I believe that this paper brings a relevant and needed contribution on the theoretical front of these questions, by paving the way to have a dynamical understanding of the static phase diagram previously obtained in ref. [5].The generic program of 1) deriving the DMFT equations in general settings, 2) solving them analytically/numerically in a specific regime such as equilibrium dynamics, is required to understand the validity of static results. The relevance of the latter for confluent biological tissues was mostly discussed in [5], hence the focus here could be directly on dynamical/algorithmic aspects, key for deep learning applications for instance.

However, before I could definitely recommend it for publication, there are some points I would like the authors to address.

———

1) I find that the abstract does not reflect the actual content of the paper, as stated in the outline provided in the introduction. In the latter, the main purpose of this work is supposedly to describe the solution of the dynamics of the model in [5], to detail how to numerically solve its equations, and to validate this approach on the specific case of gradient descent. In the abstract, it is stated that this is a study of the dynamical behaviour under generic settings, and then most of the abstract focuses on one specific result at the validation step. However, I would argue that the derivation of the DMFT equations, although an important result in itself, is only the first step towards studying the dynamics, as it provides a new tool to do so; but their analytical solution is not provided as such in this study. Furthermore, although the DMFT equations are derived in a general settings that could apply to thermal Langevin noise and active drive, here they are studied exclusively in the case of gradient descent. I would suggest a rewriting of the abstract to correct these mismatches or imprecisions.

2) In the introduction, adding some bibliography would be welcome. In its current form, the introduction describes the connection to biophysics and machine learning applications, but there are almost no associated references. Without being exhaustive, a couple of key papers and reviews are missing to provide at least a minimal background to the main biophysics motivation of this work, which is in the very title of this work. As a side remark, the papers [1-4] mentioned for vertex and Voronoi models might give the wrong impression that these models have only been recently proposed, whereas the ideas were preceding these studies (as reported in the references therein). I would suggest a slight reformulation of this passage to clarify this point and the novelty specifically brought by [1-4] on that topic.

3) At the end of the outline, it is mentioned that these DMFT equations are an intermediate case between sampling the process itself and focusing on the correlation/response function. This is a very interesting point, but if I am not mistaken, it is not mentioned again later in the paper (?). Could the authors further comment on that point after the derivation of the DMFT equation? There are now several models/problems where the DMFT equations have been derived, and highlighting such intermediate cases which can be exactly solved could enhance the impact of this study for other systems. As a side remark, the bibliography on DMFT is quite minimal and partial, so for non-experts it is not clear what has already be done and how this derivation is similar/differs from others. I would strongly suggest to expand it, including a minima the recent review by L. Cugliandolo https://arxiv.org/abs/2305.01229.

4) In the section 2.2 of the model, it would be welcome to have a schematic phase diagram of the SAT/UNSAT in (alpha,p_0) corresponding to the results of [5], to clarify which regimes are targeted via DMFT in this specific study. If I am not mistaken, such a diagram is missing in [5], but in any case the present study should be self-contained.

5) In the section 3 on DMFT, my main criticism is that in its current form I find it difficult to access to non-experts, and a bit redondant for experts. I list thereafter a couple of suggestions that I would find helpful: - Since detailed derivations have been provided elsewhere (such as for the p-spin, [12] for the continuum perceptron, and Agoritsas et al. JPhysA 52, 144002 (2019) for particle systems), I would suggest to properly rely on these previous works and to pinpoint until which point the derivations are the same, and when they start to differ. Currently there is a sentence between Eqs.(16)-(17) which states "This is the main point of departure of the present work from other closely related DMFT computations": could the authors cite the specific references, and ideally pinpoint more specifically the key equations if these papers are long? These derivation are quite technical hence this would be really helpful for the readers. - Regarding the statistical average in Eq.(8), does it also include the average over a stochastic initial condition? - In Eq. (13), shouldn’t the \mathcal{Z}{\text{dyn}} be instead Z\text{dyn}? - In Eq.(14), I would suggest to change the integration variable a so they do not collide with the ones already in Eq.(13). Also I find a bit confusing that we integrate over the gaps, while in the argument of G in Eq.(14) we have the appearance of x(a) and x(b). Could the authors explain this transition? - In Eq.(16), to highlight the dependence on the overlap, I would suggest to replace lnZ by lnZ(Q). Am I right in assuming that in Eq.(17) the expression corresponds exclusively to the contribution of lnZ(Q)? How different is Eq.(17) from alternative DMFT derivations? - Just after Eq.(13), there is the first occurrence of the "impurity problem": as such it is merely jargon, is it really necessary and/or could a comment/reference be added there? - The ansatz of Eq.(20) is quite generic for experts in Grassman algebra, but could the authors provide also the definitions of the correlation and response functions at this point? Or again point towards a reference where this is sufficiently detailed. - Later on, at Eq.(32), the indication "scalar part" might not be clear for non-experts in Grassman algebra.

6) In section 4 on the numerical integration, I understand that the equations are truncated at small dt. However, in Eq.(25) for instance, we see that we keep terms of order dt^2 and dt^3, the latter being associated to p_0^2. What happens if we truncated to dt^2 only ? Or if we try to keep an additional order? Which is the initial condition for this numerical integration? Can we consider any initial condition, for instance with a replica-symmetry-breaking starting point? When do we expect this approach to fail? In particular, how does this relate to the numerical instabilities of [29] by Manacorda and Zamponi?

7) In section 5.1, could the authors explain how they obtain the discretized dynamics of Eq.(30)? A small detail: the footnote 3 just before Eq.(30) could be confused with an exponent, which unfortunately appears on the initial spherical constraint.

8) A technical question: just after Eq.(30), the authors write "Changing sample means that we change both initial conditions and the disorder": in practice, does that mean that one uses the same seed for the random generator for the initial conditions and the disorder?

9) The validation of the DMFT numerical integration is restricted to the gradient-descent case. Could the authors comment if they expect a better or worse agreement in other cases, such as thermal Langevin noise?

10) In section 5.2, regarding the paragraph "This is not unexpected […] with gradient descent." : I do not find the statement very clear at this point, of what would be or would not be expected. Is this related to the non-commutability of the zero temperature limit and the long-time limit?

11) I do not understand the last paragraph of section 5, especially the point about decorrelation and why it is relevant or not. Would it be possible to reformulate it?

12) At last, in the section 2.1, the components of the vector \vec{x} are the degrees of freedom of the model, that could correspond to the position of the cell centers in Voronoi/vertex models, or the weights in a machine-learning setting. I understand that this identification is more at a conceptual level, to justify this simple model. However, I was wondering if the dimensionality of the components x_i could be tuned in this model, if it would still be possible to derive and address these DMFT equations if the x_i would be for instance d-dimensional vectors or matrices, and if we would expect qualitative differences on the results of this model. This issue would of course be beyond the scope of this work, but any statement that could already be done at this stage would be worth mentioning.

  • validity: -
  • significance: -
  • originality: -
  • clarity: -
  • formatting: -
  • grammar: -

Author:  Pierfrancesco Urbani  on 2023-09-26  [id 4011]

(in reply to Report 2 on 2023-09-15)

We thank the reviewer for the comments, criticism and suggestions. We detail here a reply to the points raised in the report and the changes made in the revised version of the manuscript. We follow the enumeration of the report.

1) In the new version of the manuscript we have rewritten the abstract to make it a closer description of the content of the paper.

2) We added additional references on Vertex and Voronoi models. We also included some key references on the Machine Learning side that were missing.

3) Concerning the structure of the DMFT equations. There was a comment just below Eq. (16) about the fact that the DMFT we get represents an intermediate level of difficulty with respect to more standard cases. We have included now a figure to explain the general structure of DMFT and extended the text to include a discussion on this point. Concerning the bibliography on DMFT: we added the reference to Cugliandolo’s review which contains an extended bibliography as suggested by the referee.

4) Concerning the phase diagram of the model: we have provided the relevant data for the phase transitions (the zero temperature RSB and the thermodynamic jamming point as well for alpha=1/4). We find that there is no special reason to include a full phase diagram changing alpha. This also goes beyond the scope of the manuscript that is to provide the DMFT equations and an algorithm to integrate them, as the reviewer has remarked when discussing the abstract of the manuscript. On the other hand, Reviewer 1 has asked to provide more data on the algorithmic (gradient descent) jamming transition which we added in Fig. 5.

5) We have - emphasized that our derivation closely follows another work, Agoritsas et al, J Phys. A 2017, on the spherical perceptron model. The main point of departure from this work is explicitly stated after Eq.(16). - We corrected Eq.(13) - We underlined after Eq.(14) that ${\cal Z}$ is a function of $\underline x(a)\cdot \underline x(b)$. We hope that this is clearer now. - We added the explicit dependence of the local partition function from the overlap. - We removed the jargon “impurity problem”. - We added the definition of correlation and response function. - We changed Eq. (32) by removing “Scalar part” and replacing it by saying that we are setting to zero the Grassman coordinates.

6) - Concerning the $dt$ factor in the discretized equations. We did not truncate the equations to order $dt^2$. The orders in $dt$ can be reconstructed from the equations in the continuous time limit. For each integral, there is a $dt$ factor since integrals are discretized into sums with finite timestep equal to $dt$. There is an additional $dt$ factor which comes from the fact that the gradient descent dynamics is discretized via the Euler scheme. - Concerning the initial conditions: we considered random initial conditions on the hypersphere, and we emphasized this point in the revised version. The formalism can be certainly extended to consider more complex initial conditions thermalized from the Gibbs measure in a RSB phase or at finite temperature. - Concerning instabilities and failures. We do not see contexts where our approach is unstable. The DMFT equations in continuous time are exact. The discrete time equations are approximated due to the spherical constraint but apart from this approximation, the numerical procedure to integrate them does not suffer any instability as in Manacorda & Zamponi, since the equations are not self-consistent and there are no memory kernels to converge.

7) Eq. (30) is simply a discrete time gradient descent step followed by a projection on the sphere. Indeed the numerator of the fraction is just the discrete version of gradient descent and the denominator and the term $\sqrt{N}$ are just needed to project the result on the sphere. We also corrected the position of the footnote.

8) We use just one seed in our simulations. Each time we run a new sample, we extract a new initial configuration and set of matrices $J^\mu$. We use the same pseudo-random number generator along the full simulation in order to get random samples.

9) We did not try a systematic study because when turning on thermal or active noise, one needs to average also over it. This is quite costly: the numerical simulations are heavy here since one has $O(N)$ random matrices of size $N\times N$. Anyway we do not have any a priori reason to expect a problem in the agreement when the noise is on.

10) We have changed the discussion about this point and rewritten the paragraph to clarify the fact that there is no contradiction between having RSB in the SAT phase and the gradient descent effectively finding solutions.

11) If the correlation function goes to zero, it means that the system is ending on local minima that are completely orthogonal (at leading order in N on the sphere) to the initial condition and therefore the system has no memory of its initial state. Conversely, if $C(t\to \infty,0)$ stays larger than zero the system has a non-vanishing memory of the initial condition. This gives an idea on how the energy landscape is explored.

12) We believe that doing DMFT for a real Voronoi model in $d$ dimensions is rather difficult. We do not have any particular comment on that.

---

## Round 3 · Referee Report · Anonymous (Referee 1) · 2023-10-10

Report

The authors included in Fig. 5 the location of the SAT/UNSAT transition and I think that this contribution significantly enhanced the paper.
Furthermore, the analysis of the timestep discretization is minimal but solid.
My main conceptual concerns are about the dimensionality issue which the authors acknowledge as an open problem; I find the sentence in the Perspectives section an honest remark.

I therefore think that the paper is now suitable for publication.

---

## Round 3 · Referee Report · Anonymous (Referee 2) · 2023-10-18

Report

I am satisfied with the reply to my comments provided by the authors. Therefore I now recommend the publication of the manuscript.

---

## Round 3 · Author Response

Dear Editor,

we would like to resubmit our manuscript for publication in SciPost Physics.
We have provided a detailed response to the reports and revised accordingly the manuscript.
Yours sincerely,

The authors.

---

## Round 3 · List of Changes

We have detailed the list of changes in the responses to the reports.

---

## Editorial Decision

published